# Molecular Epidemiology and Diversity of SARS-CoV-2 in Ethiopia, 2020–2022

**DOI:** 10.3390/genes14030705

**Published:** 2023-03-13

**Authors:** Abay Sisay, Derek Tshiabuila, Stephanie van Wyk, Abraham Tesfaye, Gerald Mboowa, Samuel O. Oyola, Sofonias Kifle Tesema, Cheryl Baxter, Darren Martin, Richard Lessells, Houriiyah Tegally, Monika Moir, Jennifer Giandhari, Sureshnee Pillay, Lavanya Singh, Yajna Ramphal, Arisha Maharaj, Yusasha Pillay, Akhil Maharaj, Yeshnee Naidoo, Upasana Ramphal, Lucious Chabuka, Eduan Wilkinson, Tulio de Oliveira, Adey Feleke Desta, James E. San

**Affiliations:** 1Department of Medical Laboratory Sciences, College of Health Sciences, Addis Ababa University, Addis Ababa P.O. Box 1176, Ethiopia; 2Department of Microbial, Cellular, and Molecular Biology, College of Natural and Computational Sciences, Addis Ababa University, Addis Ababa P.O. Box 1176, Ethiopia; adey.feleke@aau.edu.et; 3Centre for Epidemic Response and Innovation (CERI), School of Data Science and Computational Thinking, Stellenbosch University, Stellenbosch 7600, South Africa; 4Diagnostic Unit, Center for Innovative Drug Development and Therapeutic Trials for Africa, CDT-Africa, Addis Ababa P.O. Box 9086, Ethiopia; 5Institute of Pathogen Genomics, Africa Centers for Disease Control and Prevention (Africa CDC), African Union Commission, Roosevelt Street W21 K19, Addis Ababa P.O. Box 3243, Ethiopia; 6International Livestock Research Institute (ILRI), Nairobi P.O. Box 30709-00100, Kenya; 7Wellcome Centre for Infectious Diseases Research in Africa (CIDRI-Africa), Cape Town 7925, South Africa; 8Institute of Infectious Disease and Molecular Medicine, University of Cape Town, Cape Town 7925, South Africa; 9KwaZulu-Natal Research Innovation and Sequencing Platform (KRISP), Nelson R Mandela School of Medicine, University of KwaZulu-Natal, Durban 4001, South Africa; 10Department of Global Health, University of Washington, Seattle, WA 98105, USA

**Keywords:** COVID-19, molecular epidemiology, SARS-CoV-2, whole genome sequence, Ethiopia

## Abstract

Ethiopia is the second most populous country in Africa and the sixth most affected by COVID-19 on the continent. Despite having experienced five infection waves, >499,000 cases, and ~7500 COVID-19-related deaths as of January 2023, there is still no detailed genomic epidemiological report on the introduction and spread of SARS-CoV-2 in Ethiopia. In this study, we reconstructed and elucidated the COVID-19 epidemic dynamics. Specifically, we investigated the introduction, local transmission, ongoing evolution, and spread of SARS-CoV-2 during the first four infection waves using 353 high-quality near-whole genomes sampled in Ethiopia. Our results show that whereas viral introductions seeded the first wave, subsequent waves were seeded by local transmission. The B.1.480 lineage emerged in the first wave and notably remained in circulation even after the emergence of the Alpha variant. The B.1.480 was outcompeted by the Delta variant. Notably, Ethiopia’s lack of local sequencing capacity was further limited by sporadic, uneven, and insufficient sampling that limited the incorporation of genomic epidemiology in the epidemic public health response in Ethiopia. These results highlight Ethiopia’s role in SARS-CoV-2 dissemination and the urgent need for balanced, near-real-time genomic sequencing.

## 1. Introduction

Coronavirus disease 2019 (COVID-19) is an acute respiratory infection caused by the severe acute respiratory syndrome coronavirus 2 (SARS-CoV-2). Since its emergence in December 2019, the disease rapidly spread globally and has been responsible for more than 680,549,359 cases and 6.8 million deaths across the globe. As of January 2020, the World Health Organization (WHO) declared it a Public Health Emergency of International Concern [1,2].

On the African continent, more than 12.7 million confirmed cases and 258,122 deaths have been reported as of January 2023, far fewer than reported for other continents [1]. The low-reported number of infections and associated deaths in Africa are thought to reflect the relatively young population of Africa, climate conditions that are less favorable for the virus, reduced incidence of comorbidities, genetic factors coupled with immunological and socio-demographical aspects unique to Africa, and other anthropological factors [3,4,5]. However, many studies have also suggested that the relatively low observed rates could be a result of the poor documentation of the spread of SARS-CoV-2 in Africa [3,6], inadequate testing [7], and lack of or ineffective diagnostics [8].

Despite the relatively low numbers of cases and deaths reported in Africa and the major sequencing efforts, the significance of Africa in driving the evolution and global dissemination of the virus has been apparent [9]. Two of the most important variants of concern identified were first detected in Africa. The second reported variant of concern, Beta, was first detected in South Africa and has since been identified in at least 103 countries [10]. The Omicron variant, associated with the highest global transmission rates, albeit milder infections, was also first detected in Botswana and South Africa [11]. To date, Omicron has been detected in 113 countries. Shortly after its detection, sub-variants BA4 and BA5 were identified in South Africa [12]. Elsewhere on the continent, A23.1 was detected in Uganda and Rwanda [13] and B.1.525 in Nigeria [14]. In addition, A.30 was originally detected in Angola but is thought to originate in Tanzania [15]. Together, this highlights the need for concerted efforts towards continued close monitoring of the introduction, evolution, and dissemination of SARS-CoV-2 in Africa. However, there remain substantial gaps in knowledge of the epidemic dynamics across the continent. One such country is Ethiopia [9,16].

Ethiopia is a landlocked country in Eastern Africa [17,18]. Furthermore, it is one of the most populous countries on the continent and, following the global trend, has accordingly endured severe economic and healthcare burdens due to the ongoing COVID-19 pandemic [5,19]. On 13 March 2020, the Federal Ministry of Health in Ethiopia reported its first COVID-19 case in Addis Ababa. By February 2022, Ethiopia had recorded approximately 494,760 confirmed cases and 7572 deaths [1,20]. Despite various pandemic mitigation strategies implemented by the Government (Appendix A), the number of COVID-19 cases continued to rise, and constitutes one of the most severely affected countries on the African continent [21]. On 8 April 2020, the Ethiopian government declared a 5-month national state of emergency to control and contain the growth rate of the local epidemic and support mitigation mechanisms [21,22]. These emergency measures included the closing of schools and other academic institutes, restrictions on large gatherings, and the implementation of mask-wearing mandates [21]. These restrictions were complemented by encouraging social distancing and hand washing using multimedia campaigns [21,23]. 

Until now, no detailed molecular epidemiology and genetic diversity analyses of SARS-CoV-2 have been performed in Ethiopia. To address this problem, we analyzed 353 high-quality near-complete SARS-CoV-2 genomes sampled in Ethiopia against a globally representative set to characterize the introduction, transmission, and spread of SARS-CoV-2 in Ethiopia.

## 2. Materials and Methods

### 2.1. Ethical Considerations

This study was reviewed and approved by Addis Ababa University, College of Health Science Institutional Review Board (IRB) (IRB # 029/20/Lab), IRB of the Department of Medical Laboratory Sciences, Addis Ababa University (reference #-MLS/174/21), IRB office of Addis Ababa City Administration Health Bureau, AAPHREML (reference #-AAHB/4039/227), Addis Ababa University, College of Natural and Computational Science IRB (IRB #-CNCSDO/604/13/2021) and Yekatit 12 Hospital Medical College (IRB protocol #75/20). In addition, it was reviewed and approved by the Federal Democratic Republic of Ethiopia National Ethics Committee at the Ministry of Science and Higher Education (MoSHE) (IRB #021/246/820/21).

### 2.2. Study Design, Period, and Setting

Samples collected from confirmed Ethiopian SARS-CoV-2 cases recorded between July 2020 and February 2022 were used in this study. Specifically, 1300 nasopharyngeal (NP) swab specimens were collected using 2 mL of viral transport medium (VTM) (Guangdong, China, Miraclean Technology Co., Ltd., www.mantacc.com, accessed on 20 December 2022) (Appendix A) from patients in Addis Ababa, Amhara, Oromia, and the Southern Nations, Nationalities, and Peoples Regional States (provinces) of the country (Appendix A). The number of samples selected from each region was based on the epidemic distribution of the diseases across the country (Appendix A). These areas are known to have high population densities and cover almost three-quarters of the country’s inhabitants [24]. Viral detection in these areas was performed at the Ethiopian Public Health Institute (EPHI), AAPHREM center laboratory, Yekatit 12 Hospital, Arsho medical laboratory, and Addis Ababa University. These were amongst the established RT-PCR laboratories where the majority of COVID-19 testing was performed. Samples with cycle threshold (Ct) values under 30 were transported to the reference laboratory.

### 2.3. Quality Assurance

Samples were received at Arsho Advance Medical Laboratories Plc, an ISO 15189 accredited laboratory (https://enao-eth.org/arsho-advanced-medical-laboratory-2, accessed on 1 January 2023). Upon arrival, specimens were stored in a −80 °C freezer. To increase the success rate of whole genome sequencing, a subsequent RT-PCR test was performed at the reference laboratory prior to shipping [25], and only samples with confirmed Ct values under 30 were shipped to the sequencing laboratories. SARS-CoV-2 positive samples from the first three Ethiopian waves were shipped to the Centre for Epidemic Response and Innovation and KZN Research Innovation and Sequencing Platform (CERI/KRISP; located in Stellenbosch/ Durban, South Africa) and from the fourth wave (103) to International Livestock Research Institute (ILRI located in Nairobi, Kenya) for whole genome sequencing. Samples at CERI/ KRISP were sequenced between September 2021 and October 2022, whereas samples sent to ILRI were sequenced during March 2022.

### 2.4. RNA-Extraction and Next-Generation Sequencing

WGS was performed using either the Illumina or the Oxford Nanopore Technology (ONT) sequencing platforms. 

### 2.5. Sample Preparation

RNA was extracted from the samples on the automated Chemagic 360 instrument using the CMG-1049 kit (Perkin Elmer, Hamburg, Germany) and was stored at −80 °C until further use.

### 2.6. Genome Sequencing 

#### 2.6.1. Genome Sequencing Using Illumina Sequencing Technologies

##### Tiling-Based Polymerase Chain Reaction

The complementary DNA synthesis was performed using SuperScript IV reverse transcriptase (Life Technologies, Waltham, MA, USA) and random hexamer primers followed by gene-specific multiplex PCR using the protocol described previously [26]. In summary, SARS-CoV-2 whole genome amplification using multiplex PCR was performed using primers designed on Primal Scheme (http://primal.zibraproject.org/ accessed on 1 December 2022) to generate 400 base pair (bp) amplicons with 70 bp overlaps, covering the 30 kilobase SARS-CoV-2 genome. The PCR products were purified in a 1:1 ratio using AmpureXP purification beads (Beckman Coulter, High Wycombe, UK) and were quantified using the Qubit double-strand DNA (dsDNA) High Sensitivity assay kit on a Qubit 4.0 instrument (Life Technologies). Amplicon fragment sizes were estimated on the LabChip GX Touch (Perkin Elmer, Hopkinton, MA, USA) prior to library preparation.

The Nextera DNA Flex Library Prep kits (Illumina, San Diego, CA, USA) were used according to the manufacturer’s instructions. Undiluted tiling PCR amplicons were used. Briefly, the DNA was tagmented with bead-linked transposomes, and the tagmentation reaction was stopped with tagmentation stop buffer before proceeding to the post-tagmentation cleanup using the tagmentation wash buffer. This step was followed by eight cycles of amplification of the tagmented DNA with enhanced PCR mix and index adapters. The Nextera DNA CD indexes were used (Illumina). The libraries were cleaned using 0.9X sample purification beads and eluted in 32 µL resuspension buffer. Libraries were quantified using the Qubit dsDNA High Sensitivity assay kit on a Qubit 4.0 instrument (Life Technologies). The fragment sizes were analyzed using a LabChip GX Touch (Perkin Elmer), with expected fragments between 500 and 600 bp in size. Each sample library was normalized to 4 nM concentration, and the libraries were normalized and denatured with 5 µL of 0.2 N sodium acetate. The 12 pM library was spiked with 1% PhiX control (PhiX Control v3) and sequenced on an Illumina MiSeq platform (San Diego, CA, USA), using a MiSeq v2 500 cycle kit [26].

#### 2.6.2. Genome Sequencing Using Oxford Nanopore Technologies

The Midnight protocol was used for the amplification using Oxford Nanopore Technologies (ONT). The ONT Midnight Kit was utilized for ONT amplification. Libraries for next-generation whole genome sequencing on Oxford Nanopore technologies were prepared using the ONT Midnight Library Preparation Kit. The ARTIC amplicon v3 and v4 primer sets and Midnight whole genome sequencing protocols were used in this study. Amplicons generated were prepared for nanopore sequencing using the ONT Native Barcoding Expansion Kits as per the manufacturer’s guidelines. Libraries were multiplexed on FLO-MIN106 flow cells and run on the GridION X5) [27].

#### 2.6.3. Metadata Management

Clinical and demographic metadata were collected from the confirmed cases using a structured assessment tool by trained healthcare professionals at the triage of the health facilities and from the community. A medical record number (MRN) was assigned to each patient as a unique anonymous identifier. 

#### 2.6.4. Sequence Assembly, Alignment, and Phylogenetic Analysis

FASTQ sequences were assembled using Genome Detective 1.133 (http://www.genomedetective.com accessed on 20 January 2023) [28] and the Coronavirus tool [29]. The initial assemblies obtained from Genome Detective were visualized and manually curated using Geneious Prime version 2022.2.2 [30,31] to remove low-quality mutations. Nextclade was used to determine the quality control profiles of the sequences. The quality metrics used for the sequences in this study included genome coverage greater than 80%, ensuring the removal of unknown frameshifts, stop codons, and clustered mutations. Using the quality control reports generated by Nextclade, all mutations flagged as private mutations were either confirmed or resolved by manually inspecting the BAM files in Geneious Prime. Additionally, submissions with exceedingly high and missing data ((N) > 3000 Ns) were excluded from downstream analyses. Nextclade was further employed to assign clades and lineage classifications to the sequences [16,32]. Sequences that met these quality control thresholds were deposited in the GISAID (https://www.gisaid.org/ accessed on 14 December 2022) database (EPI_SET_221214bg, DOI: 10.55876/gis8.221214bg). 

To present a comprehensive analysis of the genomic epidemiology of SARS-CoV-2 in Ethiopia, the genomes generated in this study were combined with a globally representative set of genomes (n = 13,589) that were sampled during the same period. All samples from the bordering countries to Ethiopia were included in the analysis to better model the relationship between Ethiopia and its neighbors and infer cross-border transmission. This did not cause a sampling bias as these countries were already characterized by a low number of sequences. Additionally, all global sequences belonging to the B.1.480 lineage, which was predominant in Ethiopia and remained in circulation across the first three waves, were included. Together, this resulted in a dataset of 13,942 genomes. Acknowledgment was given to the sequencing laboratories that contributed to the analysis set (EPI_SET_221214bg, DOI: 10.55876/gis8.221214bg).

Sequences were aligned using NextAlign [33] to obtain a good codon alignment of the sequences. A maximum likelihood tree topology was inferred from the resulting alignment in IQTREE 2 [34] using the general time reversible model of nucleotide substitution [35]. The inferred phylogeny was used along with the associated metadata available from GISAID to map discrete geographical locations to each of the tips and infer locations for the internal nodes. This was performed using the *Mugration* package extension of TreeTime [36]. A custom Python script was then used to count the number of discrete changes occurring as we transcended the topology from the root towards the tips. In essence, this provided a crude estimate of the number and timing of viral exchanges (import–export) between Ethiopia and the rest of the world. 

#### 2.6.5. Data Analysis and Visualization 

The data were analyzed and visualized using R version 4.2.0 using available packages and custom scripts. Descriptive statistics were carried out and reported as frequencies, counts, percentages, and means for categorical variables. No statistical testing was performed. The inferred maximum likelihood phylogenetic tree was visualized using the R package ggtree version 3.0.4 [37].

## 3. Results

### 3.1. Socio-Demographic Characteristics of the Study Participants

A total of 1300 nasopharyngeal swabs were collected from four provinces in the country that were processed according to Appendix A. Briefly, we discarded 300 due to their high Ct values (i.e., >30) and other factors (Appendix A). The remaining 1000 samples were subjected to library preparation, and 808 yielded sufficient RNA for WGS following RT-PCR amplification. The WGS of these resulted in 353 SARS-CoV-2 sequences that passed bioinformatics quality controls (i.e., with genome coverage greater than 80%, no clustered mutations, and no misplaced stop codons). Using the quality control reports generated by Nextclade, all mutations flagged as private mutations were either confirmed or resolved by manually inspecting the BAM files in Geneious Prime. Of the 353 genomes obtained and analyzed in this study, specimens from 181 (51.3%) were obtained from female patients, whereas specimens from 172 (48.7%) were obtained from male patients. The patient ages ranged between 2 and 86 years old, although most of the cases were between 21- and 30-year-old patients, with an overall estimated median age of 31.45 (IQR = 26–46) years. More than 77% of patients presented with mild cases (Table 1). In terms of different sampling strategies employed, 167 were detected during community surveillance, 107 were hospital referrals of suspected cases (symptomatic but yet to be confirmed by laboratory results), and 79 were contacts of the confirmed cases. The hospital referrals included severe cases referred from the Yekatit 12 hospitals (Table 1, Appendix A). 

### 3.2. Local Epidemic Dynamics

Our findings show that as of February 2022, Ethiopia had experienced four distinct waves of COVID-19 infections (Figure 1A). The first occurred between mid-May 2020 and lasted until mid-November 2020, constituting ~6 months, resulting in >103,000 cases and >1500 deaths. The genetic composition of the SARS-CoV-2 lineages fueling this wave remains undetermined as Ethiopian genome surveillance was only implemented from the second wave onwards. This most likely reflects the absence of in-country sequencing capacity. However, as the pandemic progressed, the country increased its laboratory and bioinformatic expertise to achieve genome surveillance to allow for the informed implementation of pandemic mitigation strategies. However, we suspect that these lineages most likely reflect African and global trends where the first epidemiological waves were most likely fueled by an admixture of B lineages [38]. 

The second wave, which started at the end of January 2021 and continued until the end of June 2021, was dominated by the Alpha variant. However, other lineages were also detected during this time, including the Delta, Beta, B.1.1, and B.1.480 lineages. Despite this wave only lasting four months, >143,000 cases were reported, resulting in >2200 deaths. The third infectious wave occurred during August and November 2021, lasting four months and resulting in ~90,000 cases and >500 deaths. This was the deadliest wave that occurred at the time and was fueled by the Delta variant (Figure 1A,B). However, other lineages, such as B.1 and descending lineages, were also prevalent (Figure 2A,B). A paucity of data was observed during this wave leaving a blind spot in Ethiopian genome surveillance. These gaps in knowledge were most likely due to shortages in the reagents and consumables necessary to support effective genome surveillance. The final wave observed during this study period occurred from mid-December 2021 to mid-February 2022, resulting in >92,000 cases and >500 deaths. Comparable to African and global trends, this wave was fueled by the Omicron variant and resulted in the fewest deaths in comparison with the higher number of cases reported (Figure 1A,B). These case numbers were most likely far fewer than the truly infected individuals. This variant included the BA.1, BA.2, BA.3, BA.4, and BA.5 descendent lineages (Figure 2A,B).

Across the four waves, the peak number of new cases remained relatively consistent with the first and third waves reflecting a 7-day rolling average of approximately 1500 individuals. The second wave had a slightly higher 7-day rolling average of 2000 individuals, whereas the fourth had the highest at over 4000 individuals (Figure 1A). The number of new deaths occurring during each wave showed a similar trend to the average number of new cases with the second and third waves having the highest number of new deaths at approximately 50 individuals per day (Figure 1B). Vaccine rollout commenced in April 2021 in Ethiopia, and by February 2022, approximately 25% of the population had received at least one dose of Oxford-AstraZeneca or Sinopharm COVID-19 vaccines (Figure 1B).

### 3.3. Phylogenetic Reconstruction and Variant Detection in Ethiopia

Ancestral state reconstruction showed that viral imports into Ethiopia occurred between August 2020 and January 2021, with the majority of importation events occurring in August 2020 (n = 123), December 2020 (n = 130), and January 2021 (n = 169) (Figure 1C). In comparison, September and November 2020 corresponded with the highest numbers of inferred SARS-CoV-2 exports from Ethiopia (n = 109 and n = 137, respectively; Figure 1D and Appendix A). The majority of inferred importation events were from Eastern African (n = 324) origins, whereas most of the inferred exportation events were generally attributed to Western European and Asian origins (n = 20 and n = 141, respectively). Overall, we identified 570 importation and 446 exportation events during the study period.

Our early sequences sampled between November and December 2020 represented two lineages, B.1/B.1.1 and B.1.480, with most of the sequences classified as B.1/B.1.1. These two lineages continued to circulate throughout the first quarter of 2021 (January to March) (Figure 2A,B). 

The B.1.480 lineage was first detected in Australia. Notably, the first case in Africa was reported in Sudan, a neighboring country to Ethiopia. The lineage was, however, only detected in Ethiopia six months following its initial detection. It is also important to note that the last case attributed to the B.1.480 lineage was reported in Ethiopia. The lineage was also detected in neighboring countries such as Kenya, Somalia, Djibouti, and Guinea-Bissau. Overall, despite its persistence and dissemination in at least 25 countries, the lineage was outcompeted by the emerging variants of concern and only resulted in 472 confirmed cases globally. The A lineage was also detected in Ethiopia with cases interspersed between VOCs (Figure 2C). The cases did not cluster together on the phylogeny, suggesting multiple introductions. However, the lineage, as with other African countries, did not result in a rise of cases or fuel community transmission.

In January 2021, the first case of the Alpha variant was detected. Within one month, it had reached equal prevalence to B.1/B.1.1. The B.1.1 continued to circulate in low prevalence with cases detected until May 2021. Alpha was, however, short-lived as the Delta variant emerged. The Delta variant was first detected in February 2021 and continued circulating in low prevalence until May 2021 (Figure 2A,B). To rule out contamination as possibly a source of these early sequences, we examined the mutational patterns within these sequences, nucleotide distributions in the sorted, indexed, and mapped BAM files, and the timing of sequencing. Considering the timing of sequencing and that these samples were sequenced along with several samples that were also attributed to the Delta variant, contamination as a possible source could not be entirely ruled out. However, we note that these sequences conform to the molecular clock (Appendix A) estimations of the emergence of Delta globally estimated around 19 October 2020, with the earliest possible date estimated as 6 September 2020 (Appendix A). A similar pattern of early emergence and slow rise to dominance is observed in several countries across the continent [34] and in a more detailed study in neighboring Kenya. In May 2021, the Delta variant outcompeted all other lineages and variants in circulation to become the most prevalent variant (Appendix A). The predominant Delta variant sub-lineages were AY.120 (116 sequences, 46.4%) and B.1.617.2 (50 sequences, 20%). The Delta variant seeded the third wave in Ethiopia. 

Although no sequence data were available from September to December 2021, the majority of the cases sampled and sequenced from January 2022 were attributed to the Omicron lineage (Figure 1A and Figure 2A). Over the course of the study period, we identified 37 distinct lineages in circulation (Figure 2A,B and Appendix A). The phylogenetic reconstruction revealed close relatedness (not substantially observed levels of divergence) among sequences assigned to the same lineages. However, this observation did not hold for samples assigned to the Delta variant (Figure 2C). Sequences attributed to the Delta variant showed greater variation in sequence composition. Moreover, the higher levels of phylogenetic clusters in our sequences indicate and are associated, most likely, with more community transmission. Phylogenetic clusters and associated community transmission events were also observed for the Omicron lineage (Figure 2C). 

## 4. Discussion

In this study, we investigated the introduction of SARS-CoV-2 viral lineages, local transmission, and its evolution within Ethiopia. This study represents the first of its kind as no other molecular epidemiology investigation has been conducted for this country and provided an overview of the pandemic progression relative to a global context. Our results show that SARS-CoV-2 was introduced into Ethiopia on multiple occasions, with most introductions originating from Eastern African countries. These introductions included 37 distinct SARS-CoV-2 lineages and four VOCs, namely Alpha, Beta, Delta, and Omicron lineages. Of the introduced lineages, the B1.480 persisted and remained in circulation from December 2020 to August 2021.

Ethiopia remained vulnerable to SARS-CoV-2 introductions. This was despite various preventive strategies such as the mandatory wearing of masks and media awareness campaigns that were adopted together with a mandatory 14-day self-quarantine regulation for all international travelers arriving in the country [1]. Our results showed that despite these preventative measures, viral introduction events continued to occur and were followed by community transmission. This, in turn, facilitated a continued increase in case numbers. This could be attributed to the partial restriction of public transport, which is an import mode of SARS-CoV-2 transmission [1,39]. During lockdowns, the operation of economic activities did not cease as they were deemed necessary to support the already fragile economic and socio-economic situation of the country. This made lockdowns ineffective in preventing local transmission and epidemic growth. By 30 April 2021, the total number of cases and deaths had surpassed 257,442 and 3688, respectively, [17,19] and has currently increased to >499,000 and 7500 deaths.

Our findings show that Ethiopia experienced poor sampling, mostly limited to Addis Ababa. Temporal sampling was also hampered by periods of limited sampling and lack of local sequencing capacity, resulting in substantial delays that, in turn, may cause the degradation of samples during transit. However, an in-depth understanding of the COVID-19 pandemic progression in Ethiopia remains of paramount importance to aid the navigation of ongoing and future pandemics within these regions [40]. Indeed, Ethiopia has had several viral outbreaks in recent history in addition to the ongoing COVID-19 pandemic. Some of the notable outbreaks include measles and mosquito-borne illnesses such as yellow fever, chikungunya, and malaria [40,41]. Thus, the scaling up of local and regional pathogen genomic sequencing capacity and a robust pathogen genomic surveillance infrastructure is required to shorten the turnaround time and allow for early public health responses [38,42]. Additionally, during the onset of the pandemic, a survey on outbreak readiness highlighted a dire need for not only pathogen surveillance in Ethiopia, but Ethiopia further scored only 52% on the Ready Score Index, which shows that the nation still has to make progress to increase case tracing and identification, increase healthcare and testing facilities, provide clear operational guidelines on preventive measures across various organizations, businesses, and community settings, as well as proactive steps to maintain life during social and economic lockdowns [19,42,43]. Ethiopia, as with many other African countries, has limited resources to deal with these outbreaks and often depends on international aid to help control and prevent these diseases [23,44].

Over the course of four waves, we estimated 570 introductions and 446 exportation events. This is consistent with the relatively small number of lineages identified during the study period, although this could also be attributed to under and sparse sampling that likely omitted lineages circulating at low prevalence. The short generation time associated with RNA viruses could also imply that viruses circulating during the period preceding the emergence of Omicron, which may have become extinct, would not have been detected. The phylogenetic inference and that of the geographical distribution of the Ethiopian emerging B.1.480 lineage suggest that Ethiopia serves as a source for viral introductions not only to neighboring countries but also further abroad. Ethiopia serves as a travel hub for this African region connecting this region in Africa to neighboring countries and further abroad. Possible disease dissemination was most likely a result of travel to and from Ethiopia via air, land, and waterways [44,45]. Ethiopia is strategically located on the Red Sea, making it an important transit point for goods moving between Africa, the Middle East, and Asia. The main international airport in Ethiopia is Addis Ababa Bole International Airport, which facilitates high travel volumes and most likely contributed to viral amplification in the country, followed by Ethiopian-originating viral exportation events. Therefore, concerted, collaborative action is imperative to achieve pandemic mitigation.

We observed a reduction in the number of new deaths reported during the fourth wave. This is consistent with the immunity-acquired protection from a large number of infections from the preceding waves and increases in the total number of vaccinated individuals. A seroprevalence study among healthcare workers revealed a seropositivity rate of 39.6% [46], whereas another study estimated the prevalence rate at over 50% [47]. However, only about 0.4% of the population has reportedly been infected with COVID-19: much fewer than the estimated seroprevalence rates estimated for the population. Therefore, taken together, these statistics suggest that case numbers may remain widely under-reported. Reduced testing further hinders effective genome surveillance as fewer samples were available to be subjected to whole genome analyses which influences the accuracy of the epidemiological conclusions drawn.

In comparison to other countries, Ethiopia has very low vaccination coverage. As of December 2021, at least 1,000,000 Ethiopians had been vaccinated. It is important to note that COVID-19 vaccines in Ethiopia were prioritized based on criteria such as occupation, age, medical condition, people that work in aggregate settings, the elderly, and people with comorbidities known to increase the risk of adverse COVID-19 outcomes [48]. Moreover, priority was given to those deemed as frontline medical staff, followed by non-frontline medical staff. This strategy was effective in slowing the evolutionary rate of the virus as SARS-CoV-2 is prone to evolution in immunosuppressed individuals experiencing chronic infection [49,50]. Vaccination has also been shown to decrease the overall SARS-CoV-2 infection rate from 9.0% (95% CrI: 8.4–9.4%) without vaccination to 4.6% (95% CrI: 4.3–5.0%) with the highest relative reduction noted among individuals aged 65 and above. Adverse outcomes, such as deaths, are reduced by approximately 69.3% (95% CrI: 65.5–73.1%) with lowered vaccination coverage [51].

Analysis of genomes and lineages across the study period highlighted the changes in the population dynamics of the viral lineages circulating in Ethiopia. The B.1 and B.1.1 lineages dominated the first wave, persisted throughout the next two waves, and were characterized by the well-known D614G mutation [52]. This mutation has been linked to increased viral replication and transmissibility [9,45,52]. Subsequent waves were dominated by the VOCs Alpha, Beta, Delta, and Omicron. These VOCs evolved to evade and overcome host immunity from infection with ancestral viruses and vaccination. By doing so, these newly emerging variants fueled renewed infectious waves and were often associated with increased transmissibility compared to their genetic predecessors [53,54,55]. The Omicron variant (specifically, lineage BA.1.1) had a reproduction number four times higher than Delta. It is considered the fifth variant of concern first detected in November/December 2021 [56,57,58].

Moreover, our results revealed that as of February 2022, Ethiopia had experienced four distinct waves of COVID-19 infections, which were most likely fueled by new VOCs. The emergence of new variants has a direct correlation and a significantly positive association with the increase in the number of cases and prevalence as it was characterized by being more infectious due to their mutations, virulence nature, and related factors that can worsen the epidemic, as also observed in other parts of the world [38,54,56,58].

One important lineage in the context of Ethiopia was the B.1.480, a distinct B lineage, which emerged during the second wave. This variant was responsible for 2.77% of the sampled cases. Notably, this variant was previously detected in passengers originating from not only Ethiopia but also those originating from European countries such as the United Kingdom and France, as well as from travelers from the United Arab Emeritus [56,57,58]. Unlike other VOCs, this lineage did not reach global transmission and has only 473 submissions deposited to the GISAID platform; however, it persisted in Ethiopian populations for extended periods. Other worrisome mutations carried by this lineage included the Spike mutation M1229I and D614G [52,53,55,59].

## 5. Conclusions

The changing nature of the pandemic over time in Ethiopia has been driven by lineage turnovers that, while dominated by VOCs and B.1.480, also include other lineages (A, B.1, and B.1.1). The results from this study show that despite pandemic mitigation strategies implemented by the country, Ethiopia remained vulnerable to viral introduction events and served as a source of viral introductions internationally. The COVID-19 pandemic is still ongoing, and the evolution of the virus is continuing to yield new variants, some of which will likely reflect altered viral behavior such as increased transmissibility, immune evasiveness, or altered pathogenicity. It is, therefore, crucial that ongoing genomic surveillance be strengthened and conducted to present real-time data generation to determine the geographic distribution of these variants and enable the implementation of appropriate disease mitigation strategies.

## Figures and Tables

**Figure 1 genes-14-00705-f001:**
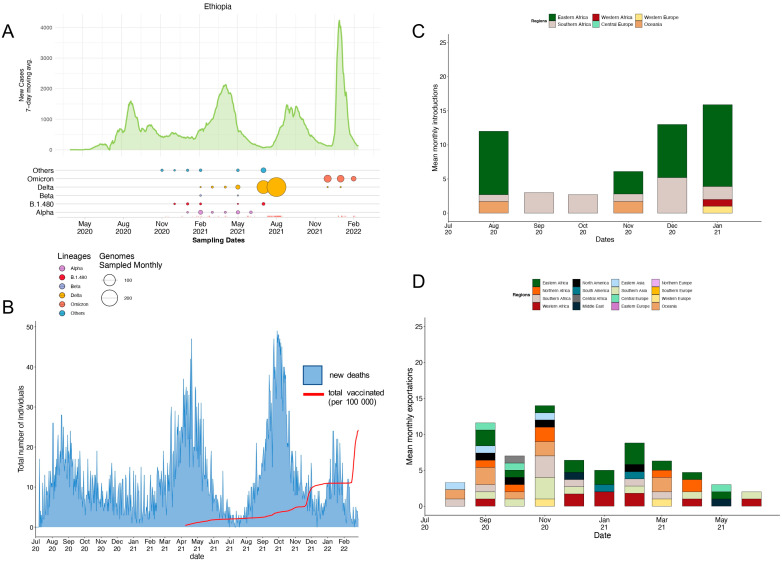
SARS-CoV-2 epidemiology and variant turnover in Ethiopia: (**A**) 7-day rolling average for the number of new SARS-CoV-2 cases. Others, represented in blue, comprise mostly B lineages, including B, B.1, and B.1.1 lineages. The size of the dot is indicative of the proportion of variant genomes sampled, whereas the rug plot (bottom segment of (**A**)) shows the distribution of genomes during the sampling period. (**B**) The proportion of individuals that have received at least one vaccine dose in Ethiopia compared to the number of new deaths that occurred. (**C**) Summary of the mean number of detected introductions into Ethiopia binned by month for different regions across the dataset. (**D**) Summary of the mean number of detected exportations out of Ethiopia to different regions binned by month.

**Figure 2 genes-14-00705-f002:**
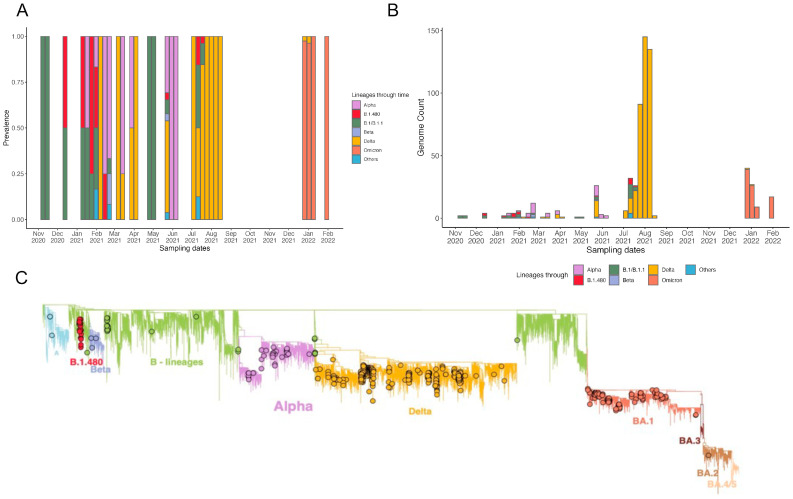
SARS-CoV-2 lineage diversity in Ethiopia. (**A**) Proportions of SARS-CoV-2 lineages, classified by Pangolin nomenclature, circulating between June 2020 and February 2022 in Ethiopia. (**B**) Absolute counts of SARS-CoV-2 genomes throughout the study period. Only VOCs and major lineages are listed in the legend. (**C**) Maximum-Likelihood phylogeny of 353 SARS-CoV-2 genomes. Branches are colored by lineage.

**Table 1 genes-14-00705-t001:** Socio-demographic characteristics of study participants along with VOCs of SARS-CoV-2 in Ethiopia, 2020–2022.

Variable	Class	Frequency (%)	Variant (VOC)			
			Alpha	Beta	Delta	Omicron
Sex	Female	181 (51.3)	10	2	99	57
	Male	172 (48.7)	7	1	102	46
Age range (years)	1–20	40 (11.3)	7	1	19	8
	21–30	121 (34.3)	1	1	80	25
	31–40	84 (23.8)	5	1	43	30
	41–50	41 (11.6)	2	0	25	12
	51–60	21 (10.2)	2	0	18	15
	>60	31 (8.8)	0	0	16	13
Reason for testing (during sampling)	Suspect	107 (30.3)	6	1	133	2
	Contacts of confirmed cases	79 (22.4)	8	2	45	1
	Community Surveillance	167 (47.3)	3	0	23	100
Clinical status of the clients while sampling	Asymptomatic *	53 (15)	3	0	37	5
	Mild	273 (77.3)	14	3	142	94
	Severe	27 (7.7)	0	0	22	4

* Asymptomatic—a person has tested positive for COVID-19 but never exhibits any signs or symptoms of the disease.

## Data Availability

All the generated sequences used for this manuscript are deposited in the GISAID sequence database and are publicly available. However, this is subject to the terms and conditions of the GISAID database, https://www.gisaid.org/ accessed on 14 December 2022, GISAID identifier: EPI_SET_221214bg, https://doi.org/10.55876/gis8.221214bg accessed on 28 January 2023.

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
