# Peer review of "Molecular Epidemiology and Diversity of SARS-CoV-2 in Ethiopia, 2020–2022"

_genes, 2023, doi:10.3390/genes14030705_

Round 1

Reviewer 1 Report

Dear Authors,

The manuscript entitled "Molecular Epidemiology and Diversity of SARS-CoV-2 in Ethiopia, 2020-2022" was reviewed.

The article is very important since it highlights on a very important topic related to molecular epidemiology of different SARS-CoV-2 variants during four different waves of COVID-19 in one of the biggest African countries "Ethiopia". 

This article has many strong points regarding the samples and the period of study in addition to the materials and methods used in this work. In parallel, it contains many weak points, mainly in the Introduction and the Results sections.

Please find below my comments and remarks concerning your work.

Minors:

01- In the whole manuscript, mainly in the introduction, authors used frequently more than one reference for each ides, even if it is a very small one. So authors are invited to remove references when it is possible (use one reference for each idea).

02- In the Supplementary materials section, authors are invited to enumerate Tables and Figures (Table S and Figure S separately: Table S1, Table S2 etc... and Figure S1, Figure S2, etc...).

03- In the Materials and Methods section, Lines 92-93, authors are invited to explain the relation between the sentence and the reference 13.

04- In the Materials and Methods section, Line 126, authors are invited to correct the reference form 18.

05- In the whole manuscript, authors are invited to explain what do they mean by information at the end of some ideas or paragraphs such as in lines 136-137 "EPI_SET_221214bg doi: 10.55876/gis8.221214bg".

06- Some part of the text need to be justified.

07- In the Discussion section, Authors are invited to use national and international references. 

08- In the Discussion section, Authors are invited to talk about the appearance of SARS-CoV-2 VOCs and the increase of new COVID-19 cases in Ethiopia. (Reference: The emergence of SARS-CoV-2 variant (s) and its impact on the prevalence of COVID-19 cases in the Nabatieh Region, Lebanon).

Majors:

01- The Introduction section is very small and weak, authors are invited to start by a small paragraph in which they represent COVID-19 and SARS-CoV-2, then in a second paragraph they can talk about SARS-CoV-2 variants and their impact on public health, mainly in increasing the transmission rate of the virus and increasing the number of COVID-19 cases.

(References: You can use other references)

i- The emergence of SARS-CoV-2 variant (s) and its impact on the prevalence of COVID-19 cases in the Nabatieh Region, Lebanon)

ii- Risk Markers of COVID-19, a Study from South-Lebanon

02- In the Figure S2, how authors said that 247 samples out the 1300 samples were excluded due to high Ct values before doing the RT-PCR!!.

03- Concerning the Materials and Methods section, the Data analysis and Visualization paragraph is very short and not well explained.

04- The Materials and Methods section, is very long, authors are invited to make it shorter with respect to all the scientific information included in this section.

05- In the Results section, First paragraph, mainly in Line 183, Authors are invited to explain a total sample size of 1300 is divided into 181 female patients and 172 male patients !!!

06- In the Results section, Kindly note that this paper lacks Figures and Tables.

Finally, and based on the large amount of major comments in your work, you have to do a lot of work to be in a better form.

Author Response

Response to reviewer comments.

Intro: 

We would like to sincerely thank the editor and the reviewers for the time taken to review our work and provide constructive feedback. Your comments have been instrumental in improving the quality of the manuscript. Below we detail how we have addressed each point as raised by the reviewers.

Reviewer 1:

The manuscript entitled "Molecular Epidemiology and Diversity of SARS-CoV-2 in Ethiopia, 2020-2022" was reviewed.

The article is very important since it highlights a very important topic related to the molecular epidemiology of different SARS-CoV-2 variants during four different waves of COVID-19 in one of the biggest African countries "Ethiopia". This article has many strong points regarding the samples and the period of study in addition to the materials and methods used in this work. In parallel, it contains many weak points, mainly in the Introduction and the Results sections.

Please find below my comments and remarks concerning your work.

Minors:

01- In the whole manuscript, mainly in the introduction, the authors frequently used more than one reference for each idea, even if it is a very small one. So authors are invited to remove references when it is possible (use one reference for each idea).

Response:

We agree with the reviewer that we often used multiple references. We carefully selected references that provide complementary perspectives to support the point being discussed. Where necessary, however, we have removed the additional references.

02- In the Supplementary materials section, authors are invited to enumerate Tables and Figures (Table S and Figure S separately: Table S1, Table S2 etc... and Figure S1, Figure S2, etc...).

Response:

We thank the reviewer for this suggestion and have enumerated the supplemental tables and figures accordingly. Please see pages 13 and 17 of the tracked change version.

03- In the Materials and Methods section, Lines 92-93, authors are invited to explain the relationship between the sentence and reference 13.

Response:

This is an important observation from the reviewer. We note that this reference was related to the text that we later revised. We have also duly revised all the references too.

04- In the Materials and Methods section, line 126, authors are invited to correct the reference form 18.

Response:

We thank the reviewer for this observation. We have added the closing brace for this reference.

05- In the whole manuscript, authors are invited to explain what do they mean by information at the end of some ideas or paragraphs such as in lines 136-137 "EPI_SET_221214bg doi: 10.55876/gis8.221214bg".

Response:

The sequence repository, GISAID, uses EPI_SET_IDs to identify a set of sequences together with the associated metadata. This allows for appropriate acknowledgement, access, and retrieval of sequences used in this study. The EPI_SET_ID provided in these sections refers to the sequences generated in this study and sequences analysed along with them as background sequences. A DOI associated with the EPI_SET is also provided for citation. We hope that this clarifies the question raised by the reviewer.

06- Some part of the text need to be justified.

Response:

All unjustified segments of text have been justified as per the recommendation of the reviewer.

07- In the Discussion section, Authors are invited to use national and international references. 

Response:

For Ethiopia, the body of work on the pandemic progression and genome surveillance remains limited. Despite the limited research conducted nationally, we set out to incorporate those available throughout the manuscript.  The authors acknowledge all relevant sources of international studies were cited and used to build our arguments.

08- In the Discussion section, Authors are invited to talk about the appearance of SARS-CoV-2 VOCs and the increase of new COVID-19 cases in Ethiopia. (Reference: The emergence of SARS-CoV-2 variant (s) and its impact on the prevalence of COVID-19 cases in the Nabatieh Region, Lebanon).

Response:

The following was added to the Results section of the manuscript (lines 301-328): Our findings show that as of February 2022, Ethiopia had experienced four distinct waves of COVID-19 infections. The first occurred between mid-May 2020 and lasted until mid-November 2020, constituting ~ 6 months, resulting in >103 000 cases and >1 500 deaths. The genetic composition of the SARS-CoV-2 lineages fuelling this wave remains undetermined as Ethiopian genome surveillance was only implemented from the second wave onwards. This most likely reflects the absence of in-country sequencing capacity. However, as the pandemic progressed the country increased it laboratory and bioinformatic expertise to achieve genome surveillance to allow for the informed implementation of pandemic mitigation strategies. However, we suspect that these lineages most likely reflect African and global trends where the first epidemiological waves were most likely fuelled by an admixture of B-lineages (34). The second wave, which started at the end of January 2021 and continued until the end of June 2021, was dominated by the Alpha variant. However other lineages were also detected during this time, including the Delta, Beta, B.1.1, and B.1.480 lineages. Despite this wave only lasting for four months, >143 000 cases were reported that resulted in >2 200 deaths. The third infectious wave occurred during August and November 2021, lasting four months and resulting in ~90 000 cases and >500 deaths. This was the deadliest wave during the pandemic progression in Ethiopia and was fuelled by the Delta variant. However, other lineages such as B.1 and descending lineages were also detected during this time. A paucity of data was observed during this wave leaving a blind spot in Ethiopian genome surveillance. These gaps in knowledge were most likely due to shortages in reagents and consumables necessary to support effective genome surveillance.  The final wave observed during this study period occurred from mid-December 2021 until mid-February 2022, resulting in >92 000 cases and >500 deaths. Comparable to African and global trends, this wave was fuelled by the Omicron variant and resulted in the fewest deaths. This variant included the BA.1, BA.2, BA.3, BA.4, and BA.5 descendent lineages.  

Majors:

01- The Introduction section is very small and weak, authors are invited to start by a small paragraph in which they represent COVID-19 and SARS-CoV-2, then in a second paragraph they can talk about SARS-CoV-2 variants and their impact on public health, mainly in increasing the transmission rate of the virus and increasing the number of COVID-19 cases.

(References: You can use other references)

i- The emergence of SARS-CoV-2 variant (s) and its impact on the prevalence of COVID-19 cases in the Nabatieh Region, Lebanon)

ii- Risk Markers of COVID-19, a Study from South-Lebanon

Response: 

We appreciate the concern of the reviewer, however, we intentionally opted for a short and concise introduction focused on Ethiopia as background and introduction to our work. The identification, global spread, and impact of SARS-CoV-2 are already very well documented and published and therefore we hope that the reviewer will agree with our view and reconsider our approach.

02- In the Figure S2, how authors said that 247 samples out the 1300 samples were excluded due to high Ct values before doing the RT-PCR!!

Response:

We agree with the views highlighted by the reviewer and thank them for pointing this out. Accordingly, to improve on the clarity, and intended meaning we have updated the figure. The samples were collected at regional testing facilities and subjected to RT-PCR. Only those with a CT value < 30 were transported to the reference laboratory for sequencing. This resulted in the discarding of 247 of the 1300 samples. Upon arrival at the sequencing laboratory, a confirmatory RT-PCR was performed prior to library prep to ensure that only samples with CT values < 30 were sequenced. This resulted in the discarding of an additional 90 samples. We hope that this is now clear in the text and the figure.

03- Concerning the Materials and Methods section, the Data analysis and Visualization paragraph is very short and not well explained.

Response:

We have revised this section to add some more detail. Specifically, we highlight the reporting of descriptive statistics and that we did not perform any statistical testing in this study. Moreover, the ggtree package is a standard tree visualization package for R.

04- The Materials and Methods section, is very long, authors are invited to make it shorter with respect to all the scientific information included in this section.

Response:

We have revised the methods section to remove any redundant information and retain all the information that is crucial for the replication of the study.

05- In the Results section, First paragraph, mainly in Line 183, Authors are invited to explain a total sample size of 1300 is divided into 181 female patients and 172 male patients !!!

Response:

We have revised this section accordingly to show the right progression of the numbers. Although 1300 samples were collected, only 353 resulted in high-quality sequences analysed in this study. The distributions presented here focused on 353. We hope that this clarifies the reviewer's concern.

06- In the Results section, Kindly note that this paper lacks Figures and Tables.

Response:

We appreciate the reviewer's concern, although it is unclear what message the reviewer intended to convey. Can the reviewer advise what additional figures and tables they would like to have added to the results beyond the two main figures and one table, and the supplementary figures? Please note the figures included in this review are Figure 1 and Figure 2, constituting seven illustrations and graphs.

Finally, and based on the large amount of major comments in your work, you have to do a lot of work to be in a better form.

Response:

We have taken the time to address the comments raised by the reviewer and incorporated their suggestions to the best of our ability. We hope that the revised version of the manuscript meets the reviewer's expectations.

Reviewer 2 Report

The manuscript entitled “Molecular Epidemiology and Diversity of SARS-CoV-2 in Ethiopia, 2020-2022” provides unknown data on the diversity of SARS-CoV-2 clades and lineages which contributed to the different waves in Ethiopia, one of the most affected country in Africa during the pandemics. Authors illustrate the need for improving molecular surveillance in Africa in order to improve variant identification, to decrease the risk of cryptic transmissions towards neighboring countries, and to guide public health decisions. Despite the little total number of sequences compared to the total number of COVID-19 cases, the authors identify different lineages which are suspected to have seeded the successive waves in Ethiopia. I strongly encourage the publication of these data. Nonetheless, the manuscript needs, in my opinion, to be revised for different major points:

Major points:

·      Methods used to obtain sequences are not clear: 

-       First, authors assembled raw sequences by using Genome Detective. According to Vilsker, et al. doi:10.1093/bioinformatics/bty695, this tool is useful for metagenomics sequencing. I just wonder if primer sequences from ARTIC protocol are removed in case of amplicon-strategy. This step is essential for not providing false sequences in these regions. Besides, please specify thresholds minimum depth for variant calling.

-       Secondly, authors firstly propose a genome coverage above 80% as quality criteria. How many sequences are below 90% coverage? Actually, the threshold of 80% is very low and lineage annotation by Nextclade should be carefully interpreted. I understand that conditions for RNA conservation were not optimal. However, under 90%, more criteria like spike coverage or Nextstrain quality controls should be discussed. Co-circulation of lineages may lead to recombinants which might be missed by a low coverage, leading to biases for downstream analyses. Moreover, the threshold of 80% does not correspond to the missing data criteria of <3000Ns (90%) proposed by authors. Please clarify the methodology and discuss the impact of sequencing quality on phylogeny and other analyses.

·      The authors explain that the B.1.480 lineage is thought to have originated in Ethiopia before disseminating in neighboring countries and in Europe. The results supporting this hypothesis are not showed. Besides, line 378 in the discussion section, authors explain that this lineage was previously identified in travelers from Europe. It is not clear if these travelers were in Ethiopia or in another country. Maybe a figure or table could better illustrate the results for this lineage.

·      The sampling strategy should be more discussed. It seems that sequences generated herein are from community patients, not from hospital. But there are also severe cases presented in the Table 1. How are defined severe cases? Authors should specify which part of the samples are based on viruses circulating in community and which part originate from hospital. 

·      Figures need to be improved:

a.     Figure 1C and Figure 1D do not correspond to what is explain in the manuscript. The mean number of monthly introductions is not the same as the total number of introduction events. Besides the gray color in Fig 1D does not represent a specific region.

b.     Figure 1A. Size of the dot is not indicative of the proportion. The legend indicates a number. Besides, to what does the term “rug” correspond in the legend? 

c.     Figure S4 needs a legend for a better understanding.

Minor Revisions:

1)    Lines 27-28 and lines 47-48 : In the introduction, number of cases and deaths are estimated on February 2022.  In the abstract, the globally “same numbers” are presented with the mention “more than” on January 2023. This is a shortcut that may not really represent the actual number of total cases, especially with the Omicron variant and subvariants with strongly increased transmissibility. Actually, with this formulation, it seems that the fifth wave was not significant in Ethiopia.

2)    Line 114 : please indicate the version of the ARTIC protocol.

3)    Lines 180-195 : The results are not correctly presented. Table 1 is based on the 353 samples, not on the 1300 nasopharyngeal swabs. The description of socio-demographic features of study participants should appear after the inclusion of good-quality sequences.

4)    Line 202 : “Each wave was characterized by higher rates of transmission and fatality than previous waves (Figure 1A)”.  Add also the Figure 1B.

5)    Line 236 : Double “only”

6)    Line 256 : clade instead of lineage ?

7)    Line 258 : “Moreover, higher levels of phylogenetic clusters, associated community transmission, were observed during the study period” This sentence is not clear. Are these observations based on Figure 2C and Secondary cases of the Table 1? Or maybe authors are meaning : higher number of clustered sequences?  

8)    Line 344 : The fourth wave was due to Omicron variant. Several papers also confirmed the relative decreased virulence of this variant.

9)    Lines 385-395 : these lines should appear in the results section.

10) Line 399 : The A lineage is never presented in the results or the Figures, but appear in conclusion

11) Table S1 : Column “Remark” is not filled. Please harmonize the presentation and sentences.

12) Reference 45 has been published in Clinical Infectious Disease

Author Response

We would like to sincerely thank the editor and the reviewers for the time taken to review our work and provide constructive feedback. Your comments have been instrumental in improving the quality of the manuscript. Below we detail how we have addressed each point as raised by the reviewers.

Reviewer2:

The manuscript entitled “Molecular Epidemiology and Diversity of SARS-CoV-2 in Ethiopia, 2020-2022” provides unknown data on the diversity of SARS-CoV-2 clades and lineages which contributed to the different waves in Ethiopia, one of the most affected country in Africa during the pandemics. Authors illustrate the need for improving molecular surveillance in Africa in order to improve variant identification, to decrease the risk of cryptic transmissions towards neighboring countries, and to guide public health decisions. Despite the little total number of sequences compared to the total number of COVID-19 cases, the authors identify different lineages which are suspected to have seeded the successive waves in Ethiopia. I strongly encourage the publication of these data. Nonetheless, the manuscript needs, in my opinion, to be revised for different major points:

Major points:

  •     Methods used to obtain sequences are not clear: 

-       First, authors assembled raw sequences by using Genome Detective. According to Vilsker, et al. doi:10.1093/bioinformatics/bty695, this tool is useful for metagenomics sequencing. I just wonder if primer sequences from ARTIC protocol are removed in case of amplicon-strategy. This step is essential for not providing false sequences in these regions. Besides, please specify thresholds minimum depth for variant calling.

Response:

Yes, Genome Detective employs a metagenomics approach for the identification and discovery of viruses (and lately bacteria). Each identified virus is then processed to provide a final consensus sequence. The platform has been used throughout the SARS-CoV-2 pandemic for genome assembly including for the detection of Beta, Omicron and Omicron variants BA4/5 all published in high-impact journals (PMIDS: 36108049, 35760080, 35042229, 33690265) cited in nearly 200 publications to date. The performance of SARS-CoV-2 genome assembly has been tested against the arctic SARS-CoV-2 bioinformatics protocol and the results were confirmed to be consistent and often better. A threshold of 20 is used for the detection of consensus mutations and where we are not confident, as stated in the manuscript, we manually curated these sequences by inspecting checked the bam files in Geneious Prime to confirm the mutations. Genome detective also checks for and trims the primer sequences if any were detected. 

-       Secondly, authors firstly propose a genome coverage above 80% as quality criteria. How many sequences are below 90% coverage? Actually, the threshold of 80% is very low and lineage annotation by Nextclade should be carefully interpreted. I understand that conditions for RNA conservation were not optimal. However, under 90%, more criteria like spike coverage or Nextstrain quality controls should be discussed. Co-circulation of lineages may lead to recombinants which might be missed by a low coverage, leading to biases for downstream analyses. Moreover, the threshold of 80% does not correspond to the missing data criteria of <3000Ns (90%) proposed by authors. Please clarify the methodology and discuss the impact of sequencing quality on phylogeny and other analyses.

Response:

We agree with the reviewer's concern regarding sequences with less than 90% coverage, however, coverage of 80% is the standard that we have used all through the pandemic. The data presented in this manuscript also does not contain clear traces of any viral recombination. Pangolin and Nextclade assigned the correct lineage or variant and were both in agreement. We are therefore confident that although some sequences fell below 90%, our results are robust to support the claims detailed in this manuscript.

  •     The authors explain that the B.1.480 lineage is thought to have originated in Ethiopia before disseminating in neighboring countries and in Europe. The results supporting this hypothesis are not showed. Besides, line 378 in the discussion section, authors explain that this lineage was previously identified in travelers from Europe. It is not clear if these travelers were in Ethiopia or in another country. Maybe a figure or table could better illustrate the results for this lineage.

Response:

Our preliminary analysis suggested that B.1.480 originated from Ethiopia however considering that subsequent data submitted to GISAID refutes this, we have removed this statement from the results and only focus on its persistence over time within the population. 

  •     The sampling strategy should be more discussed. It seems that sequences generated herein are from community patients, not from hospital. But there are also severe cases presented in the Table 1. How are defined severe cases? Authors should specify which part of the samples are based on viruses circulating in community and which part originate from hospital. 

Response:

We thank the reviewer for their insight on this matter you. As indicated in the 'Study Design, Period, and Setting' section of the materials and methods, multiple sampling strategies were employed including community surveillance (n=170), confirmed cases (n=78), and suspected cases (symptomatic but yet-to-be-confirmed by laboratory results) (n=105)” collected from four provinces in the country part of routine genomic surveillance of SARS-CoV-2 throughout the pandemic. The severe cases were among the confirmed cases tested at Yekatit 12 hospital.

  •     Figures need to be improved:
  1.     Figure 1C and Figure 1D do not correspond to what is explain in the manuscript. The mean number of monthly introductions is not the same as the total number of introduction events. Besides the gray color in Fig 1D does not represent a specific region.

Response: 

To ensure the robustness of our ancestral state reconstruction, we run our analysis in 10 replicates. The values were plotted to represent the mean values across the replicates. We have accordingly updated the rest of the manuscript to show the mean values instead of the totals.  The grey color represents Central Africa

  1.     Figure 1A. Size of the dot is not indicative of the proportion. The legend indicates a number. Besides, to what does the term “rug” correspond in the legend? 

Response: 

We agree with the reviewer that the dot is determined by the count of genomes. However, the reviewer should note that dot size scales relative to the total number of genomes hence a proportion. We have updated the figure to remove the "rug" from the legend. The rug plot (bottom segment of Figure 1A) shows the distribution of genomes during the sampling period. We have updated the legend accordingly.

  1.     Figure S4 needs a legend for a better understanding.

Response:

We have updated the legend as follows;

"Chord plot showing the exchange of SARS-CoV-2 viruses between Ethiopia and other countries as inferred by import and export analysis for the period between June 2020 and February 2022. The flat end from Ethiopia denotes viral exports from Ethiopia, while the pointed ends toward Ethiopia denote viral imports into Ethiopia. The scale represents mean import/exports as inferred against 10 replicates."

Minor Revisions:

1)    Lines 27-28 and lines 47-48 : In the introduction, number of cases and deaths are estimated on February 2022.  In the abstract, the globally “same numbers” are presented with the mention “more than” on January 2023. This is a shortcut that may not really represent the actual number of total cases, especially with the Omicron variant and subvariants with strongly increased transmissibility. Actually, with this formulation, it seems that the fifth wave was not significant in Ethiopia.

Response: 

The review makes a valid point. For this analysis, we used publicly available data from https://covid19.who.int/region/afro/country/et. On February 2022 there were 468,495 documented cases and 7265 deaths whereas, as of January 2023 just over 499 000 cases, and 7 500 COVID-19-related deaths had been documented. It is true that the difference highlights potential under-reporting, consistent with the fact that there has been a general reduction in testing globally. Still the results show a significant impact on the population in Ethiopia. We have updated the discussion to also highlight this. 

2)    Line 114 : please indicate the version of the ARTIC protocol.

Response:

Line 161-162: The ARTIC amplicon v3 and v4 primer sets and Midnight whole genome sequencing protocols were used in this study.

3)    Lines 180-195 : The results are not correctly presented. Table 1 is based on the 353 samples, not on the 1300 nasopharyngeal swabs. The description of socio-demographic features of study participants should appear after the inclusion of good-quality sequences.

Response:

Although samples were collected from 1300 participants, after sample QC, library prep, sequencing and sequence QC, only 353 sequences had successfully been sequenced and passed QC. Table 1 details the characteristics of the 353. We have improved the text to clarify that.

4)    Line 202: “Each wave was characterized by higher rates of transmission and fatality than previous waves (Figure 1A)”.  Add also Figure 1B.

Response:

We thank the reviewer for this suggestion, we have added Figure 1B. 

5)    Line 236: Double “only”

Response:

We have identified and removed the second, "only".

6)    Line 256: clade instead of lineage?

Response:

We appreciate the thinking of the reviewer, however, we chose to use the term lineages in line with the naming convention of the sequence clusters i.e. Pango lineages.

7)    Line 258 : “Moreover, higher levels of phylogenetic clusters, associated community transmission, were observed during the study period” This sentence is not clear. Are these observations based on Figure 2C and Secondary cases of the Table 1? Or maybe authors are meaning : higher number of clustered sequences?  

Response:

Thank you. Revised the sentence.

8)    Line 344 : The fourth wave was due to Omicron variant. Several papers also confirmed the relative decreased virulence of this variant.

Response:

We thank the reviewer for pointing this out and we have removed this sentence.

9)    Lines 385-395 : these lines should appear in the results section.

Response: 

We have incorporated the lines into the results as advised by the reviewer.

10) Line 399 : The A lineage is never presented in the results or the Figures, but appear in conclusion

Response: 

We have added the lineage to the results section on page 8, lines 328-333 as follows.

"The A lineage was also detected in Ethiopia with cases interspersed between VOCs (Figure 2C). The cases did not cluster together on the phylogeny, suggesting multiple introductions. Yet, the lineage, as with other African countries did not result in a rise of cases or fuel community transmission."

11) Table S1: Column “Remark” is not filled. Please harmonize the presentation and sentences.

Response:

Thank you. We have removed the column.

12) Reference 45 has been published in Clinical Infectious Disease

Response:

We have updated the reference to the published version.

Round 2

Reviewer 1 Report

Dear Authors,

Your paper was re-reviewed,

The new version is better than the first draft, thank you for the modifications you made,

Unfortunately some remarks and comments were not modified correctly by the authors, therefore I would like to highlight again on some majors in your work that must be taken into consideration:

01- In the Introduction section, authors are invited to add more information in this section, they must enriched the introduction.

02- In the Results section, Kindly note that this paper lacks Figures and Tables. (You mention in your paper: Although no sequence data was available from September to December 2021, the majority 414 of the cases sampled and sequenced from January 2022 were attributed to the Omicron 415 lineage (Fig 1A & 2A). Over the course of the study period, we identified 37 distinct 416 lineages in circulation (Fig 2 A, Fig 2B... ) but you do not have Fig 1A nor Fig 2A etc... You have supplementary figures that can be mentioned as (Fig S...)

03- In the Discussion section, authors are invited to add international references to the manuscript since most references are from studies in Ethiopia.

04- In the Discussion section, authors are invited to talk about the impact of the appearance of new variants on the transmission rate (Ref:  The emergence of SARS-CoV-2 variant (s) and its impact on the prevalence of COVID-19 cases in the Nabatieh Region, Lebanon). (Ref: Risk Markers of COVID-19, a Study from South-Lebanon)

Best Regards,

Author Response

We would like to sincerely thank the reviewer for the time taken to review our work and provide constructive feedback. Your comments have been instrumental in improving the quality of the manuscript. Below we detail how we have addressed each point as raised by the reviewers

01- In the Introduction section, authors are invited to add more information in this section, they must enriched the introduction.

Response:

The following three paragraphs are now added to the revised manuscript: line 39-68

The Coronavirus disease 2019 (COVID-19) is an acute respiratory infection caused by the severe acute respiratory syndrome coronavirus 2 (SARS-CoV-2). Since its emergence in December 2019, the disease rapidly spread globally and has been responsible for more than 680,549,359 cases and 6.8 million deaths across the globe. As of January 2020, the World Health Organization (WHO) had declared it a Public Health Emergency of International Concern [1, 2].

On the African continent, more than 12.7 million confirmed cases and 258,122 deaths have been reported as of January 2023, far fewer than reported for other continents [1]. These low-reported number of infections and associated deaths in Africa are thought to reflect the relatively young population of Africa, climate conditions that are less favorable for the virus, reduced incidence of comorbidities, genetic factors coupled with immunological and socio-demographical aspects unique to Africa, and other anthropological factors [3-5]. However, many studies have also suggested that the relatively low observed rates could be a result of the poor documentation of the spread of SARS-CoV-2 in Africa [3, 6], inadequate testing [7], and lack of or ineffective diagnostics [8].

Despite the relatively low numbers of cases and deaths reported in Africa and major sequencing efforts, the significance of Africa in driving the evolution and global dissemination of the virus has been apparent [9]. Two of the most important variants of concern identified were first detected in Africa. The second reported variant of concern, Beta, was first detected in South Africa and has since been identified in at least 103 countries [10]. The Omicron variant, associated with the highest global transmission rates, albeit milder infections was also first detected in Botswana and South Africa [11]. To date, Omicron has been detected in 113 countries. Shortly after its detection sub-variants BA4 and BA5 were identified in South Africa [12]. Elsewhere on the continent, the A23.1 was detected in Uganda and Rwanda and B.1.581 in Nigeria [13]. Together, this highlights the need for concerted efforts towards continued close monitoring of the introduction, evolution, and dissemination of SARS-CoV-2 in Africa. Yet, there remain substantial gaps in knowledge of the epidemic dynamics across the continent. One such country is Ethiopia [9, 14].

02- In the Results section, Kindly note that this paper lacks Figures and Tables. (You mention in your paper: Although no sequence data was available from September to December 2021, the majority 414 of the cases sampled and sequenced from January 2022 were attributed to the Omicron 415 lineage (Fig 1A & 2A). Over the course of the study period, we identified 37 distinct 416 lineages in circulation (Fig 2 A, Fig 2B... ) but you do not have Fig 1A nor Fig 2A etc... You have supplementary figures that can be mentioned as (Fig S...)

Response:

Our manuscript findings have been supported by the two main figures and one table, and the supplementary tables and figures. That was submitted and included for the reviewers. We have reincorporated it in the revised version, please do contact the editor to assist in retrieving the figures supporting this manuscript as this may be accessed using another link provided.

03- In the Discussion section, authors are invited to add international references to the manuscript since most references are from studies in Ethiopia.

Response:

In our discussion, we have used a total of 26 references to discourse our findings, of these only 7 were from Ethiopia. As this manuscript explores the COVID-19 pandemic in Ethiopia we feel that this is sufficient. However, following the reviewer’s suggestion, in addition to these, we add some international references to confer and enrich our discussion to place our findings in a global context. Thanks.

04- In the Discussion section, authors are invited to talk about the impact of the appearance of new variants on the transmission rate (Ref:  The emergence of SARS-CoV-2 variant (s) and its impact on the prevalence of COVID-19 cases in the Nabatieh Region, Lebanon). (Ref: Risk Markers of COVID-19, a Study from South-Lebanon)

Response:

 We thank the reviewer for their input. Accordingly, we included the following paragraph in the revised manuscript, at line 451-456.

“Moreover, our results revealed that as of February 2022, Ethiopia had experienced four distinct waves of COVID-19 infections which were most likely fuelled by new VOCs.  The emerging of new variants has a direct correlation and a significantly positive association with the surge and increase in the number of cases and prevalence as it was characterized by being more infectious due to their mutations, virulence nature, and related factors that can worsen the epidemic, as also observed in other parts of the world [38, 54, 56, 58]”

Reviewer 2 Report

The new manuscript has been clearly improved. I appreciate the clear responses provided by the authors.

I have just minor requests :

Study Design : What is the difference between "community surveillance" and "contact of confirmed cases" ? Were contact cases diagnosed at hospital ? This would be interesting to show on the Figure 2C the different populations. Nosocomial events are also a part of SARS-CoV-2 contaminations. Authors suggest numerous phylogenetic clusters associated with community transmission. How is this stated ?

In the first paragraph added in the results section : "WGS of these resulted in 353 SARS-CoV-2 sequences that passed bioinformatics quality controls. (i.e., with genome coverage greater than 80%, less than 60 private mutations, no clustered mutations, and no misplaced stop codons)". On what is based the threshold of 60 private mutations ? Not mentionned in Material and Methods.

In the paragraph added in Local Epidemic Dynamic section : the numbers of deaths (>500 for the delta wave and >500 for the Omicron wave) do not reflect the suggestion of « deadliest wave » and « fewest deaths ».

Author Response

We would like to sincerely thank the reviewer for the time taken to review our work and provide constructive feedback. Your comments have been instrumental in improving the quality of the manuscript. Below we detail how we have addressed each point as raised by the reviewers

Study Design: What is the difference between "community surveillance" and "contact of confirmed cases" ? Were contact cases diagnosed at hospital ? This would be interesting to show on the Figure 2C the different populations. Nosocomial events are also a part of SARS-CoV-2 contaminations. Authors suggest numerous phylogenetic clusters associated with community transmission. How is this stated?

Response:

Contact of confirmed cases occurs when  a person has been in contact with a person with laboratory-confirmed COVID-19 disease less than 48 hours before the onset of their first symptoms) and they are then subsequently contacted and tested.  This information (contact history with confirmed COVID-19 patients) was gained using the assessment tool while our data and samples were collected at designated testing sites and done at the COVID-19 laboratory. Whereas, “community surveillance” was deemed a type of case finding and identification of COVID-19 cases through the aid of public health professionals and conducting a screening using a systematic manner in the community.

We appreciate the reviewer’s suggestion. However, this message is already illustrated in table 1, and for avoiding duplication of messages and that going to possibly boring our esteemed fellow readers. Thus, we believe to leave it, since overpopulated and duplicated information makes the figure dreary at this stage. Hope you understand us, as we all rely on all figures to make and convey a clear message, easy to read, and informative to catch the reader’s attention

We thank the reviewer for their input. We accordingly revised the sentences “…Moreover, the higher levels of phylogenetic clusters in our sequences indicate and are associated, most likely, with more …”, on lines 349-351

In the first paragraph added in the results section : "WGS of these resulted in 353 SARS-CoV-2 sequences that passed bioinformatics quality controls. (i.e., with genome coverage greater than 80%, less than 60 private mutations, no clustered mutations, and no misplaced stop codons)". On what is based the threshold of 60 private mutations?

Response:

We agree with the views of the reviewer and that this sentence is misleading. In the materials and methods section, we note the use of Nextclade. Nextclade is an open-source software for SARS-CoV-2 genomic analysis that allows for the rapid analysis of viral genome sequences. It includes a module for flagging high levels of private mutations, which are mutations that differ between the query sequence and the nearest neighbor sequence on a reference tree that is representative of the global phylogeny... Normally, a pretty stringent but acceptable threshold for private mutations is at least 10 or less. For sequences with higher numbers of private mutations, as reported by Nextclade, we manually inspected the BAM files and resolved these mutations.

We have revised the sentence to state: “Using the quality control reports generated by Nextclade, all mutations flagged as private mutations were either confirmed or resolved by manually inspecting the BAM files in Geneious Prime, line 182-184 and 225- 227.”

In the paragraph added in Local Epidemic Dynamic section : the numbers of deaths (>500 for the delta wave and >500 for the Omicron wave) do not reflect the suggestion of « deadliest wave » and « fewest deaths ».

Response:

We thank you and appreciate the reviewer’s insight. We accordingly updated the paragraph, lines 264- 281.